# An Empirical Study of Instruction-tuning
# Large Language Models in Chinese

**Qingyi Si**[1,2]*, **Tong Wang**[1,2]*, **Zheng Lin**[1,2]†
**Xu Zhang**[3], **Yanan Cao**[1,2], **Weiping Wang**[1]

[1]Institute of Information Engineering, Chinese Academy of Sciences, Beijing, China
[2]School of Cyber Security, University of Chinese Academy of Sciences, Beijing, China
[3]APUS AiLMe Lab

{siqingyi,wangtong1997,linzheng,caoyanan,wangweiping}@iie.ac.cn,
zhangxu@apusai.com

## Abstract

The success of ChatGPT validates the potential of large language models (LLMs) in artificial general intelligence (AGI). Subsequently, the release of LLMs has sparked the open-source community's interest in instruction-tuning, which is deemed to accelerate ChatGPT's replication process. However, research on instruction-tuning LLMs in Chinese, the world's most spoken language, is still in its early stages. Therefore, this paper makes an in-depth empirical study of instruction-tuning LLMs in Chinese, which can serve as a cookbook that provides valuable findings for effectively customizing LLMs that can better respond to Chinese instructions. Specifically, we systematically explore the impact of LLM bases, parameter-efficient methods, instruction data types, which are the three most important elements for instruction-tuning. Besides, we also conduct experiment to study the impact of other factors, e.g., chain-of-thought data and human-value alignment. We hope that this empirical study can make a modest contribution to the open Chinese version of ChatGPT. This paper will release a powerful Chinese LLM that is comparable to ChatGLM. The code and data are available at https://github.com/PhoebusSi/Alpaca-CoT.

## 1 Introduction

The emergence of ChatGPT gives humanity a real sense of hope for AGI for the first time, and inspires researchers to realize the importance of LLM research. However, the closed source of LLMs (e.g., GPT-3 (Brown et al., 2020) and PaLM (Chowdhery et al., 2022)) coupled with the requirement for massive computing resources to build the exclusive LLM has deterred researchers from reaching the LLM training stage. Subsequently, a series of "API research" based on GPT-3 and ChatGPT are con-

stantly emerging, which stimulate the specific capabilities of frozen LLMs (e.g., Chain-of-Thought (Wei et al., 2023; Wang et al., 2023; Kojima et al., 2023)) or guide them to complete specific tasks (Yang et al., 2022; Shen et al., 2023), by calling OpenAI interfaces and carefully designing prompts without model training.

The unexpected disclosure of the pre-trained LLaMA (Touvron et al., 2023) model changes this situation, and has sparked a surge of excitement in the LLM research community. This is the first open LLM with competitive performance. Recently, Alpaca (Taori et al., 2023) uses self-instruct (Ouyang et al., 2022) and ChatGPT to generate 52K instructions, which can enable LLaMA to respond to various human instructions like ChatGPT. This open project verifies the important role of instruction-tuning (Wei et al., 2022; Chung et al., 2022) open LLMs in replicating the ChatGPT process.

Given the open LLM LLaMA and Alpaca's high-quality instruction data, there is still a challenge for researchers: even the instruction-tuning of the 7B model still requires high computational resources. To address this problem, Alpaca-LoRA extends the parameter-efficient method LoRA to LLaMA, which further reduces the computing cost of instruction-tuning. It further sparks extensive research in the open-source community on instruction-tuning for LLMs. On this basis, more LLMs (e.g, Bloom (Workshop, 2023), GPT-J (Wang and Komatsuzaki, 2021)) are shown to have significant improvements in instruction-following performance with instruction-tuning. On the other hand, more instruction data is constantly being proposed, e.g., Belle (Ji et al., 2023) constructs Chinese instructions in the same way, and ShareGPT collects a large number of real human-ChatGPT conversations.

However, research on instruction-tuning LLMs in Chinese, the world's most spoken language, is still in its early stages. LLM bases, parameter-

---

*Equal contribution.
†Corresponding author: Zheng Lin.

| base LLMs | Tokens | Language | Size |
|---|---|---|---|
| LLaMA | 1T | Mainly in English | 6.7B, 33B, 65B |
| Bloom | 341B | 46 languages | 7.1B, 176B |
| moss-base | 700B | Chinese and English | 16.1B |
| ChatGLM* | 1T | Chinese and English | 6B |

| sft LLMs | Base | Instruction data | |
|---|---|---|---|
| Vicuna | LLaMA | 70k human-ChatGPT conversations | |
| Bloomz & Bloomz-mt | Bloom | 13-crosslingual-task mixture xP3 & xP3mt | |
| moss-sft | moss-base | moss-002-sft-data | |
| ChatGLM | ChatGLM* | *unknowable* | |

Table 1: Pre-training details of the popular open base LLMs (upper). Supervised fine-tuning (sft) details of the open sft LLMs. "*" denotes closed source, which denotes that ChatGLM only releases the supervised fine-tuned version. More details of each open LLM can be found at Appendix A.2.

efficient methods, and instruction data are three essential elements for customizing Chinese ChatGPT-like LLMs. There are no tutorials in the academic community on them yet. Some important questions have not yet been explored and answered: 1) "*Which open LLM is more suitable as a foundation for Chinese instruction-tuning?*", 2) "*How do parameter-efficient methods other than LoRA affect LLMs?*" and 3) "*What is the impact of various types of instruction datasets?*" To answer these questions, we collect a range of LLMs, parameter-efficient methods, and instruction datasets. Besides, we consider the AGI (instruction-following) capability and professional knowledge reserve (human exams) of models, and correspondingly select two benchmarks Belle-eval (Ji et al., 2023) and MMCU (Zeng, 2023) for comprehensively evaluation.

We also conduct experiments to explore several other factors that may affect the final performance. Specifically, we find that tuning with Chain-of-Thought (CoT) data can improve the ability to respond to complex reasoning questions. Different LLMs may be suitable for different language prompts (excluding instruction parts) in instruction-tuning. Human-value alignment results into slight performance drop. On the basis of the above findings, this paper carefully instruction-tunes a powerful Chinese LLMs that is comparable to ChatGLM.

The contributions can be summarized as follows: (1) We are the first to systematically study on instruction-tuning in Chinese through adequate experiments, which can serve as a cookbook that provides valuable findings for customizing Chinese version of ChatGPT. (2) We release a powerful Chinese LLM that is comparable to ChatGLM.

## 2 Instruction-tuning Triplets

### 2.1 Preliminaries

**Problem Formulation.** LLM bases $\mathbf{m} \in M$, parameter-efficient methods $\mathbf{p} \in P$ and instruction datasets $\mathbf{d} \in D$ are the three crucial elements in instruction-tuning. This section examines the impact of each element in the instruction-tuning triplet $(\mathbf{m}, \mathbf{p}, \mathbf{d})$ on final performance. We traverse the target element to thoroughly explore its impact, while fixing two elements in the triplet to control the variable. For example, we analyze the impact of different types of instruction datasets by comparing the performance of $\{(\mathbf{m}, \mathbf{p}, \mathbf{d}_i)\}_1^{|D|}$.

**Benchmarks.** We select two evaluation benchmarks, Belle-eval and MMCU, to comprehensively evaluate LLM competencies in Chinese. Belle-eval is constructed by self-instruct with ChatGPT, which has 1,000 diverse instructions that involve 10 categories covering common NLP tasks (e.g., QA) and challenging tasks (e.g., code and math). We use ChatGPT to rate the model responses based on the golden answers. This benchmark is considered to be as the assessment of AGI (instruction-following) capability. MMCU is a collection of Chinese multiple choice questions in four professional disciplines of medicine, law, psychology and education (e.g., Gaokao examination). It allows LLMs to take exams in human society in a multiple-choice test manner, making it suitable for evaluating the breadth and depth of knowledge of LLMs across multiple disciplines. More statistics and details are shown in Appendix A.1.

### 2.2 Open Large Language Models

To answer "*Which open LLM is more suitable as a foundation for Chinese instruction-tuning?*", we

| | LLMs | Code | Open QA | Brain Storm | Clf. | Math | Gen. | Sum. | Rewrite | Close QA | Extract | Avg. |
|---|---|---|---|---|---|---|---|---|---|---|---|---|
| *base* | LLaMA | 45.0 | 6.6 | 17.9 | 41.3 | 16.8 | 40.2 | 42.5 | 61.2 | 28.8 | 27.6 | 32.8 |
| | Bloom | 51.1 | 15.4 | 41.8 | 56.4 | 26.0 | 53.7 | **63.3** | 74.1 | 42.5 | **58.0** | 48.2 |
| | moss-base | 40.3 | 5.8 | 52.9 | 20.4 | 13.1 | 51.4 | 32.8 | 47.0 | 4.0 | 17.6 | 28.5 |
| *sft* | Vicuna | 62.6 | 17.8 | 84.6 | 48.4 | 34.7 | 85.0 | 59.7 | 77.2 | 39.4 | 40.5 | 55.0 |
| | Bloomz | 49.7 | 15.5 | 54.4 | 52.2 | 15.5 | 60.9 | 37.5 | 71.0 | 43.8 | 38.4 | 43.9 |
| | Bloomz-mt | 46.8 | 15.2 | 58.5 | 49.8 | 15.1 | 59.1 | 45.0 | 72.4 | 40.8 | 33.8 | 43.7 |
| | moss-sft | 63.9 | 25.7 | 78.5 | 33.4 | 15.1 | 78.4 | 46.0 | 58.5 | 22.7 | 22.4 | 44.5 |
| | ChatGLM | 64.7 | 39.9 | **91.8** | 53.2 | 46.5 | 91.0 | 61.9 | 82.4 | **48.8** | _53.8_ | **63.4** |
| | **Ours** | **72.4** | **41.4** | _91.5_ | **64.7** | _36.1_ | **92.3** | _62.5_ | **85.8** | _45.6_ | 38.9 | _63.1_ |
| | ChatGPT | 84.3 | 54.9 | 93.0 | 74.6 | 88.2 | 94.4 | 64.0 | 87.2 | 66.9 | 58.1 | 76.6 |

Table 2: Performance of open LLMs on Belle-eval. "Ours" is our carefully designed instruction-tuned LLM, which is discussed in detail in Section 4. The best scores are bold, the second best scores are underlined. The results of ChatGPT are only for display and will not be compared.

| LLMs | Med. | Psyc. | Law | Edu. | Avg. |
|---|---|---|---|---|---|
| LLaMA | 2.66 | 3.75 | 1.14 | 1.95 | 2.38 |
| Bloom | 4.29 | 4.50 | 5.68 | 1.41 | 3.97 |
| moss-base | 7.70 | 7.70 | 7.98 | 8.47 | 7.96 |
| Vicuna | 10.50 | 10.20 | 8.44 | 13.12 | 10.57 |
| Bloomz | **35.15** | **32.30** | **17.29** | **36.87** | **30.40** |
| Bloomz-mt | _33.77_ | _31.40_ | 15.59 | _35.12_ | _28.97_ |
| moss-sft | 16.78 | 14.45 | 9.47 | 15.43 | 14.03 |
| ChatGLM | 31.04 | 28.65 | _15.86_ | 29.96 | 26.38 |
| **Ours** | 27.88 | 23.60 | 13.86 | 25.73 | 22.77 |
| ChatGPT | 50.90 | 43.50 | 23.98 | 45.72 | 41.03 |

Table 3: Performance of open LLMs on MMCU.

collect and evaluate the most widely used open LLMs[1] available in the open source community, as shown in Table 1.

### 2.2.1 Evaluation of Existing LLMs

**Performance on Belle-eval.** Table 2 shows the scores of open LLMs on Belle-eval. The upper part shows base LLMs while the lower part shows the the supervised fine-tuned (sft) LLMs. We can derive several observations: 1) For base LLMs, Bloom performs the best because its ROOTS (Workshop, 2023) pre-training dataset has a large proportion in Chinese (261B), second only to English. Although moss-base is an LLM specifically proposed for Chinese, its performance is poor as it is obtained through further pre-training based on the CodeGen (Nijkamp et al., 2023) model and has only seen 100B Chinese data. 2) For sft LLMs, ChatGLM outperforms others by large margins, thanks to the fact that it is trained with the most

Chinese tokens and HFRL. 3) The Open QA, Math, CloseQA and Extract categories are still very challenging for existing open LLMs. 4) Vicuna and moss-sft have clear improvements compared to their bases, LLaMA and moss-base, respectively. The gain brought by Vicuna is more significant (22.2%) because its instruction data, collected by ShareGPT, are the real conversations between humans and ChatGPT, with higher quality. 5) In contrast, the performance of sft models, Bloomz and Bloomz-mt, is reduced compared to the base model Bloom, because they tend to generate a shorter response (refer to Appendix C.1). Unfortunately, ChatGPT often scores lower for short responses. The reason for this phenomenon is that Bloomz and Bloomz-mt are fine-tuned from xP3, which is built from the NLP task collection where many tasks have brief annotations.

**Performance on MMCU.** Table 3 shows the accuracy of LLMs on MMCU, we find that: 1) All base LLMs perform poorly because it is almost difficult to generate content in the specified format before fine-tuning, e.g., outputting option numbers. 2) All sft LLMs outperform their corresponding base LLMs, respectively. In particular, Bloomz performs the best (even beats ChatGLM) because it can generate option number directly as required without generating other irrelevant content (refer to Appendix C.2), which is also due to the data characteristics of its supervised fine-tuning dataset xP3. 3) Among the four disciplines, law is the most challenging for LLMs.

The LLMs' performance on MMCU is much lower than that of Belle-eval because MMCU requires higher professional knowledge. All open

---

[1]We mainly explore the version around 7b of them (except for 16b of Moss), which strike a balance between performance and computation resource requirements.

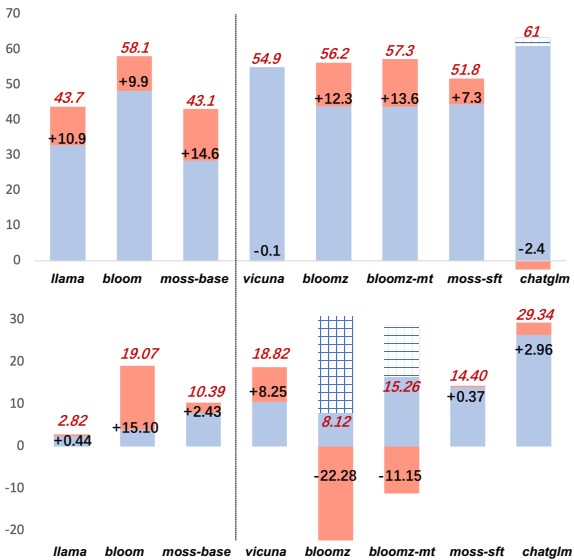

Figure 1: Performance gains (denoted by origin bars) of open LLMs on Belle-eval (Upper) and MMCU (Lower) from instruction-tuning. The instruction-tuned performance is denoted by blue bars and red numbers.

| LLMs | Param. | Layer | Belle | MMCU |
|------|--------|-------|-------|------|
| AdaLoRA | 5.6M | each | 51.9 | 14.28 |
| LoRA | 15M | each | 58.1 | 19.07 |
| prompt | 0.08M | embed | 46.8 | 4.49 |
| p-tuning | 1.1M | embed | 46.0 | 15.50 |
| prefix | 30.8M | each | 51.6 | 16.18 |
| SadapterP | 7.5M | each | 56.9 | 17.24 |
| SadapterH | 15M | each | **58.7** | **20.23** |
| P-adapter | 15M | each | 55.7 | 15.47 |
| SadapterP-l | 60M | each | 55.0 | 16.10 |
| SadapterH-l | 120M | each | 54.7 | 18.60 |
| P-adapter-l | 120M | each | 56.3 | 19.40 |

Table 4: Comparison of parameter-efficient methods. "Param." denotes the trainable parameter quantity. "Layer" denotes the layers adapters are added. "-l" denotes the version with large number of parameter. More details of each parameter-efficient methods can be found in Appendix A.3.

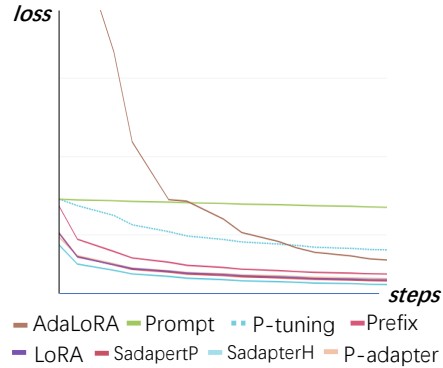

Figure 2: Training loss over steps for different parameter-efficient methods.

LLMs still have significant room for improvement compared to ChatGPT.

### 2.2.2 Instruction-tuning Different LLMs

To determine the appropriateness of different LLMs as a foundation for instruction-tuning in Chinese, we fine-tune all the open LLMs with the same parameter-efficient method LoRA and the same instruction dataset Alpaca-GPT4. The results are shown in Figure 1[2], where we find that: 1) On Belle-eval, the performance improvement of sft LLMs brought by instruction-tuning is not as significant as that of base LLMs, except for sft Bloomz and Bloomz-mt. This is because the instructions of xP3 used for their supervised fine-tuning are not diverse enough. 2) Vicuna and ChatGLM encounter performance drops after instruction-tuning, because Vicuna is trained from real human-ChatGPT conversations, with better quality than Alpaca-GPT4. ChatGLM adopts HFRL (Ouyang et al., 2022), which may be no longer suitable for further instruction-tuning. 3) On MMCU, most LLMs achieve performance boosts after instruction-tuning, with the exception of Bloomz and Bloomz-mt, which have unexpectedly significantly decreased performance. This is because that original Bloomz and Bloomz-mt excel in multiple choice questions, but after further instruction-tuning, they suffer catastrophic forgetting.

After instruction-tuning, Bloom has significant improvements and performs well on both benchmarks. Although ChatGLM beats Bloom consistently, it suffers performance drop during instruction-tuning. Therefore, among all open LLMs, Bloom is most suitable as a foundation model in the subsequent experiments for Chinese instruction-tuning exploration.

### 2.3 Parameter-efficient Methods

For most researchers, parameter-efficient methods are essential for instruction-tuning due to limitations in computing resources. These methods tend to freeze the pre-trained model weights and injects trainable weights (adapters), which greatly reduces the number of trainable parameters. To answer "*How do parameter-efficient methods other than LoRA affect LLMs?*", we collect a range of parameter-efficient methods to instruction-tune

---
[2]Full results are shown in Table 10 and 11 in App. B.1.

| Datasets | Num | Con | Type | Source |
|----------|-----|-----|------|--------|
| Aplaca-GPT4 | 49K | SI | diverse instructions | GPT-4 |
| Belle | 1.54M | SI | diverse instructions | text-davinci-003 |
| ShareGPT-zh | 158K | MIX | human-ChatGPT conversations | human & ChatGPT |
| moss-sft-data | 1.76M | SI | diverse instructions | text-davinci-003 |
| instinwild | 52K | SI | diverse instructions | text-davinci-003 |
| firefly | 1.65M | COL | 23 NLP tasks (1.15M) & Belle (0.5M) | human |
| HC3 | 40K | MIX | QA dataset collection | human & ChatGPT |
| xP3/zh | 1.07M | COL | 16 NLP tasks | human |
| COIG-ccmc | 68K | MIX | LLM-LLM role-playing chats based on a knowledge graph dataset | |
| COIG-trans | 66K | COL | 2000+ NLP tasks | translated |
| COIG-exam | 64K | COL | examinations in China | human |
| pCLUE | 1.2M | COL | 9 NLP tasks | human |

Table 5: The details of existing Chinese instruction datasets. "Con" column shows the dataset construction methods. "Type" column shows the data types. "Source" column shows the source where the data was generated. "SI" and "COL" denotes the self-instruct methods and the collection of existing datasets, respectively. "MIX" denotes the joint construction of humans and machines. "translated" denotes a translation from non-Chinese instructions. We filtered all datasets to remove incomplete instructions. More details of each dataset can be found in Appendix A.4.

Bloom on Alpaca-GPT4 dataset.

**Comparison of Parameter-efficient Methods.** From Table 4, several observations can be derived: 1) SadapterH performs the best among all parameter-efficient methods, which can be used as an alternative to LoRA. 2) P-tuning and prompt-tuning underperform others by large margins, indicating that only adding trainable layers in the embedding layer are not enough to support LLMs for generation tasks. 3) Although AdaLoRA is an improvement of LoRA, its performance has a clear drop, possibly because the LoRA's trainable parameters for LLMs are not suitable for further reduction. 4) Comparing the upper and lower parts, it can be seen that increasing the number of trainable parameters for sequential adapters (i.e., SadapterP and SadapterH) does not bring gain, while the opposite phenomenon is observed for parallel adapters (i.e., P-adapter). This may provide inspiration for the design of adapters for LLM. Since LoRA is currently the most popular parameter-efficient method, if not otherwise specified, we adopt LoRA by default in the experiments.

**Training Loss.** Figure 2 shows the training loss of different parameter-efficient methods. We find that: 1) Prompt-tuning and P-tuning converge the slowest and has the highest losses after convergence. This shows that embedding-only adapters are not suitable for instruction-tuning LLMs. 2) The initial loss of AdaLoRA is very high because it requires simultaneous learning of parameter budget allocation, which makes the model unable to fit the training data well. 3) The other methods can quickly converge on training data and fit it well.

## 2.4 Chinese Instructions Datasets

Alpaca (Taori et al., 2023) inspires researchers to further explore instruction data. To systematically explore "*What is the impact of various types of instruction datasets?*", we gather popular open Chinese instructions (as shown in Table 5) to fine-tune Bloom with LoRA.

**Performance on Belle-eval.** As shown in Table 6 upper part, it can be seen that: 1) the instruction data constructed by ChatGPT (e.g., using self-instruction methods or collecting real human-ChatGPT conversations) consistently enhances the instruction-following ability with 3.1 ∼ 11-point score increases. 2) Among these datasets, Belle has the best performance due to the largest amount of instruction data. However, the performance of models trained on moss-sft-data, containing more data built in a similar way, is unsatisfactory. This is because moss-sft-data's instructions sacrifice the diversity to achieve the goals of helpfulness, honey, and harmlessness. 3) The performance brought by the Alpaca-GPT4 instructions is the second best, with only 49K being comparable to the 1.54M Belle. This is because Alpaca-GPT4 uses the GPT-4 engine while Belle uses the text-avinci-003 engine, which further illustrates that improving data quality can reduce the demand for data volumes. 4) Instinwild brings the least performance gains among them because the seed instructions it crawls from Tweet ("in wild") are not as comprehensive as those (like Alpaca) carefully designed by hu-

| Datasets | Code | Open QA | Brain Storm | Clf. | Math | Gen. | Sum. | Rewrite | Close QA | Extract | Avg. |
|---|---|---|---|---|---|---|---|---|---|---|---|
| – | 51.1 | 15.4 | 41.8 | 56.4 | 26.0 | 53.7 | 63.3 | 74.1 | 42.5 | 58.0 | 48.2 |
| Alpaca-GPT4 | 53.7 | 35.0 | **88.2** | 50.5 | 36.8 | **89.5** | 60.3 | 82.9 | 40.0 | 44.1 | 58.1 |
| Belle | **64.3** | **37.3** | 86.0 | **66.2** | 22.3 | 88.6 | 62.6 | **83.5** | 43.3 | 38.1 | **59.2** |
| ShareGPT-zh | 53.7 | 25.4 | 75.6 | 47.8 | 34.0 | 81.2 | 62.8 | 80.8 | 33.5 | 41.6 | 53.6 |
| moss-sft-data | 55.4 | 21.7 | 81.6 | 51.7 | 24.3 | 81.8 | **65.4** | 78.1 | 37.6 | **45.1** | 54.3 |
| instinwild | 55.5 | 24.5 | 70.3 | 40.8 | 26.9 | 79.6 | 61.8 | 78.4 | 37.7 | 37.0 | 51.3 |
| firefly | 61.4 | 31.8 | 79.8 | 53.2 | 26.5 | 84.4 | 62.5 | 83.1 | 36.7 | 47.8 | 56.7 |
| HC3 | 46.8 | 21.3 | 58.2 | 24.6 | 33.9 | 44.4 | 30.3 | 47.8 | 36.5 | 19.7 | 36.4 |
| xP3/zh | 32.1 | 13.3 | 17.0 | 39.4 | 12.1 | 24.1 | 21.5 | 66.5 | 40.0 | 28.1 | 29.4 |
| COIG-trans | 34.7 | 15.5 | 58.9 | 47.9 | 24.4 | 62.1 | 51.5 | 80.0 | 37.7 | 44.6 | 45.7 |
| COIG-ccmc | 39.2 | 15.0 | 44.0 | 16.5 | 21.2 | 22.3 | 23.3 | 31.6 | 19.8 | 19.3 | 25.2 |
| COIG-exam | 24.2 | 15.4 | 53.7 | 33.3 | 17.9 | 60.2 | 44.0 | 69.6 | 27.1 | 24.9 | 37.0 |
| pCLUE | 19.7 | 19.1 | 53.5 | 34.3 | **36.9** | 50.5 | 34.0 | 68.5 | **44.4** | 39.3 | 40.0 |

Table 6: Belle-eval performance of models instruction-tuned from Bloom on different instruction datasets.

| Datasets | Med. | Psyc. | Law | Edu. | Avg. |
|---|---|---|---|---|---|
| – | 4.29 | 4.50 | 5.68 | 1.41 | 3.97 |
| Alpaca-GPT4 | 27.70 | 17.35 | **17.59** | 13.63 | 19.07 |
| Belle | 21.57 | 19.05 | 15.13 | 15.28 | 17.76 |
| ShareGPT-zh | 8.83 | 8.75 | 12.18 | 12.04 | 10.45 |
| moss-sft-data | 16.60 | 17.75 | 11.72 | 17.29 | 15.84 |
| instinwild | 14.90 | 17.45 | 11.50 | 14.17 | 14.51 |
| firefly | 22.49 | 18.30 | 9.69 | 20.50 | 17.75 |
| HC3 | 9.15 | 14.10 | 8.01 | 7.24 | 9.63 |
| xP3/zh | 20.43 | 19.50 | 15.62 | 19.60 | 18.79 |
| COIG-trans | 18.62 | 17.9 | 11.31 | 16.51 | 16.09 |
| COIG-ccmc | 7.31 | 10.15 | 8.12 | 7.63 | 8.30 |
| COIG-exam | **32.56** | **26.90** | 16.18 | - | - |
| pCLUE | 20.29 | 25.40 | 13.91 | **27.80** | **21.85** |

Table 7: MMCU performance of models instruction-tuned from Bloom on different instruction datasets.

mans. 5) These ChatGPT-based data mainly have a significant improvement effect on open generation tasks such as Brain Storm and Generation, while there is a significant decrease in tasks that require high reading comprehension skills, such as Close QA and Extract, which require completing tasks based on given materials. This inspires researchers to consider the reading-comprehension ability for building more comprehensive instruction datasets.

The lower part of Table 6 shows the results of models trained on dataset-based data, which is mainly constructed by collecting NLP or examination datasets. These instruction datasets cause damage to the model's instruction-following ability, because the form and intent of each NLP or examination dataset are unitary, which can easily be overfitted. Among them, COIG-trans performs the best because it involves over 2000 different tasks with a wide variety of task instructions. In contrast, xP3[3] and COIG-ccmc have the worst negative impact on model performance. Both of them only cover few types of tasks (translation and QA for the former, counterfactual correction conversations for the latter), which hardly cover the popular instructions and tasks for humans.

**Performance on MMCU.** Table 7 compares the performance on MMCU brought by different instruction datasets. 1) Instruction-tuning on each dataset can always result in performance improvement. 2) Among the ChatGPT-based data shown in the upper part, ShareGPT-zh underperforms others by large margins. This may be due to the fact that real users rarely ask multiple choice questions about academic topics. 3) Among the dataset-collection data shown in the lower part, HC3 and COIG-ccmc results in the lowest accuracy because that the unique questions of HC3 is only 13K, and the task format of COIG-ccmc is significantly different with MMCU. 4) COIG-exam[4] brings the greatest accuracy improvement, benefiting from the similar task format as MMCU.

## 3 Other Important Factors

**Problem Formulation.** In addition to the essential three elements ($\mathbf{m}, \mathbf{p}, \mathbf{d}$) discussed above, there are many factors worth exploring, e.g., CoT. If not otherwise specified, we use Bloom as the LLM base, LoRA as the parameter-efficient method, and Alpaca-GPT4 as the instruction data. On this basis,

---

[3]We use 1/3 of its Chinese data due to its large quantity.
[4]Due to the overlap between COIG-exam and MMCU-Edu., the accuracy on Edu. discipline will not be reported.

| Data | Belle-eval | | | MMCU | |
|---|---|---|---|---|---|
| | Code | Math | Avg. | Edu. | Avg. |
| Alpaca-GPT4 | 53.7 | 36.8 | **58.1** | 13.63 | 19.07 |
| Alpaca-GPT4+CoT | 60.8 | **41.7** | 57.9 | 21.56 | 19.85 |
| Alpaca-GPT4+CoT* | **62.9** | 39.5 | 57.2 | **22.07** | **21.56** |

Table 8: The impact of chain-of-thought data on complex tasks requiring reasoning. "*" denotes that using prompt " 先思考，再决定 " ("think step by step" in Chinese) during inference.

we explore its impact by observing the performance changes after considering the target factor.

**Chain-of-Thought Data.** Chain-of-Thought is a hot topic in LLM research. Existing works find that adding rationales or explanations to the inference prompts (Wei et al., 2023; Wang et al., 2023; Kojima et al., 2023) (based on APIs of GPT-3 and ChatGPT) or training corpus (Wei et al., 2022; Chung et al., 2022; Zhang et al., 2023c) (based on normal language models, e.g, T5(Raffel et al., 2020) and FLAN-T5(Wei et al., 2022)) can enhance the model's reasoning ability, which is useful for solving complex problems. However, extending CoT into Open LLM has not yet been thoroughly explored. Alpaca-CoT (Qingyi Si, 2023) uses several qualitative examples to demonstrate the effectiveness of CoT in reasoning. A systematic evaluation is still necessary. To this end, this paper conducts experiments to analyze the impact of CoT data for LLMs.

We collect 9 CoT datasets and their prompts from FLAN (Wei et al., 2022), and then translates them into Chinese using Google Translate. We compare the performance before and after adding CoT data during instruction-tuning in Table 8. "Alpaca-GPT4+CoT" outperforms "AlpacaGPT4" in the Code and Math tasks that require strong reasoning ability. Besides, there is also a significant improvement in MMCU education task, which is derived from the questions of Gaokao, involving a range of subjects, e.g., math, physics, history. The accuracy improvement across all subjects illustrates that the CoT reasoning ability is generally required in various subjects. However, CoT training data cannot continue to bring benefits to all tasks, and on the contrary, it will cause slight performance degradation on more tasks. The full results can be found in Appendix B.3.

Inspired by (Kojima et al., 2023), we add a sentence" 先思考，再决定 " ("think step by step" in Chinese) at the end of each instruction, to induce

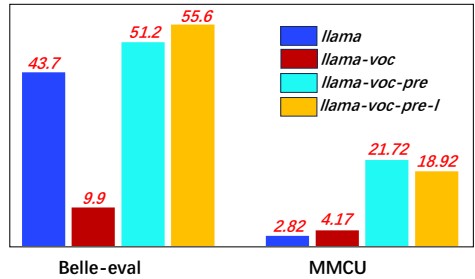

Figure 3: Performance comparison of LLaMA and its expanded vocabulary versions. "llama-voc", "llama-voc-pre" and "llama-voc-pre-l" denotes instruction-tuning the models obtained by further pre-training LLaMA with a expanded vocabulary on 0B, 20B and 100B Chinese tokens, respectively.

the model to respond to instructions based on the chain-of-thought. As shown in the line of "Alpaca-GPT4+CoT*", the simple sentence can further improve the performance of reasoning tasks Code and Education, while the Math performance is slightly inferior to "Alpaca-GPT4+CoT". This may require us to further explore more robust prompts.

**Expansion of Chinese Vocabulary.** Intuitively, the number of Chinese tokens in the tokenizer's vocabulary affects LLMs' ability to express Chinese. For example, if a Chinese character is in the vocabulary, it can be represented by a single token, otherwise it may require multiple tokens to represent it. Because Bloom adopts a vocabulary of 250k tokens which cover most Chinese characters, we mainly conduct experiments on LLaMA, which uses SentencePiece (Sennrich et al., 2016; Kudo and Richardson, 2018) (32K vocabulary size) covering few Chinese characters.

As shown in Figure 3, we find that the performance of "llama-voc" is severely inferior to "llama" on Belle-eval, and is almost unable to respond correctly to MMCU's instruction. This indicates that it is not feasible to perform instruction-tuning without pre-training on vast data. This is because the embedding corresponding to the newly added Chinese token are random and meaningless, which results in the model being unable to understand the meaning of the instructions.

To make the newly added Chinese token meaningful, Cui et al. uses 20B and 100B token Chinese corpus to further pre-train LLaMA and obtain "llama-voc-pre" and "llama-voc-pre-l" models. We use Alpaca-GPT4 to instruction-tune these models, and find that, pre-training on more Chinese cor-

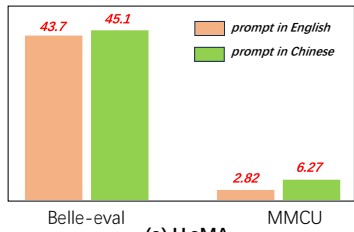
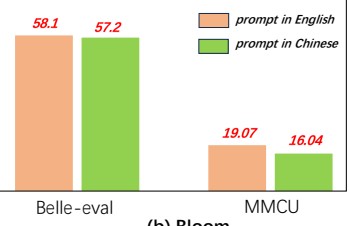
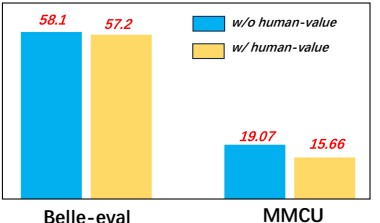

Figure 4: Performance comparison of instruction-tuning with prompts in English and Chinese. The specific prompts used in our experiments can be found in Appendix D.2.

Figure 5: Performance comparison of instruction-tuning with and without human-value alignment.

pus with expansion of Chinese vocabulary are consistently helpful for instruction-following ability. Counterintuitively, "llama-voc-pre-l" is inferior to "llama-voc-pre" on MMCU shows that pre-training on more data may not necessarily lead to higher performance for academic exams.

**The Languages of Prompts.** The popular open instruction-tuned LLMs, e.g., Alpaca and Vicuna, tend to uses prompts in English. One intuitive question is, *Is instruction-tuning in Chinese more suitable for using Chinese prompts?* Figure 4 shows the results of using Chinese and English prompts based on LLaMA and Bloom. When instruction-tuning LLaMA, using Chinese prompts can improve the performance on both benchmarks compared to English prompts, while we observe the opposite phenomenon on Bloom. This demonstrates that using Chinese prompts for models with weaker Chinese abilities (e.g., LLaMA) can effectively help respond in Chinese, while for models with good Chinese abilities (e.g., Bloom), using prompts in English (the language they are better at) can better guide the model to understand the process of fine-tuning with instructions.

**Human Value Alignment.** To avoid LLMs generating toxic content, aligning them with human values is a crucial issue. We add the human-value alignment data built by COIG (see App. A.4 for details) into the instruction-tuning to explore its impact. Figure 5 compares the results of instruction-tuning with and without human-value alignment, which shows that the human-value alignment results into a slight performance drop. How to balance the harmlessness and performance of LLMs is a research direction worth exploring in the future.

## 4 Towards a Better Chinese LLM

**Problem Formulation.** The goal of this section is to find a optimal triplet $(\mathbf{m}, \mathbf{p}, \mathbf{d})$ that maximizes the comprehensive capabilities:

$$\max_{\mathbf{m},\mathbf{p},\mathbf{d}} \sum_{t \in T} \left( \mathcal{E}_t \left( f_\mathbf{d} \left( \mathbf{m}, \mathbf{p} \right) \right) \right) \quad (1)$$

where $\mathcal{E}_t$ denotes the evaluation of every generative ability $t$ from both Belle-eval and MMCU $T$, $f_d(\mathbf{m}, \mathbf{p})$ denotes the model obtained by instruction-tuning frozen LLM $\mathbf{m}$ with parameter-efficient method $\mathbf{p}$ on instruction dataset $\mathbf{d}$.

**Our Instruction-tuned LLM.** On the basis of the findings above, we carefully design the instruction-tuning process and publicly release a Bloom-based high-performance LLM, which is comparable to ChatGLM and far surpassing Moss. In particular, we select a dataset combination with significant gains on Belle-eval or MMCU to improve our model's comprehensive ability. Besides, we carefully design a suitable prompt to induce our model for better-quality generation. The implementation details can be found in Appendix D.1.

As shown in Table 2, our model is superior or comparable to ChatGLM in most categories on Belle-eval, except for the challenging Math and Extract tasks. Besides, our model slightly underperforms ChatGLM on MMCU and outperforms other LLMs that do well in Belle-eval by clear margins. It is worth emphasizing that our model has much fewer trainable parameters (16M) based on LoRA than that of ChatGLM adopting full parameter fine-tuning (6B).

## 5 Conclusion

This paper is the first to conduct a thorough empirical study on instruction-tuning open large language models in Chinese, with a detail discussion of a

range of large language models, parameter-efficient methods, and Chinese instruction datasets. In addition, we explore several other important factors, including CoT, vocabulary, language of prompts and human-value alignment. Based on the empirical exploration, we publicly release a LLM, that is rival to ChatGLM, with detailed implementation details.

## Limitations

Most experimental results are based on parameter-efficient methods, which may differ from the results of full parameter fine-tuning. However, we believe that the findings and conclusions in this paper are still applicable for full parameter fine-tuning. In addition, instruction-tuning based on parameter-efficient methods has broader application and research scenarios.

## Ethics Statement

The open LLMs used in this paper may be driven by certain biases in their training data, and pose a risk of toxic generation. There may also exist harmful stereotypes in the open instruction datasets we are discussing. There is still a long way to explore the safety of LLMs.

## Acknowledgement

This work was supported by National Natural Science Foundation of China (No. 61976207) and National Social Science Foundation of China (No. 21AZD145).

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

| Belle-eval | | MMCU |
|---|---|---|
| Code | Open QA | Medicine |
| 38 | 285 | 2819 |
| Brainstorm | Classification | Psychology |
| 179 | 65 | 2000 |
| Math | Generation | Law |
| 75 | 98 | 3695 |
| Summary | Rewrite | Education |
| 40 | 131 | 3331 |
| Close QA | Extract | - |
| 52 | 37 | - |
| Total | 1000 | 11845 |

Table 9: Data statistics of Belle-eval and MMCU.

# A More Details about the Work Involved

## A.1 Benchmarks

**Belle-eval** During this evaluation, ChatGPT is used to rate (from 0 to 1) the model response based on the ground truth answer. A score of 0 indicates that the model response is completely unacceptable, while a score of 1 indicates that the response perfectly solves the input instruction. The prompts and instructions for samples in each category are rich and varied. We consider the capability examined in this dataset to be AGI (instrcution-following) capability.

**MMCU** MMCU (Zeng, 2023) is collected from online public resources, covering 11845 multiple choice questions in four professional disciplines. There are several subtasks under education and medicine disciplines. The average accuracy of all subtasks is considered the discipline score. Only when a generated answer and the annotated ground truth option number or option content completely match, is the answer considered correct. This evaluation is relatively rigid for expected outputs. We consider the capability examined in this dataset as the reserve of professional knowledge (to deal with human examinations).

These two assessments complement each other to some extent. Table 9 shows the data statistics of these two benchmarks.

## A.2 Open Large Language Models

### A.2.1 Base LLMs

**LLaMA** LLaMA (Touvron et al., 2023) is a decoder-only language model based on the Transformer (Vaswani et al., 2017) architecture, and is trained on more tokens (1T, 1.4T) than what is typically used (Hoffmann et al., 2022). It ranges from

7B to 65B parameters and outperforms existing LLMs (e.g., GPT-3 (Brown et al., 2020) and PaLM (Chowdhery et al., 2022)) with fewer parameter magnitudes. However, the vocabulary of its tokenizer contains fewer Chinese characters, which affects its expressive power in Chinese.

**Bloom** Bloom (Workshop, 2023) is a multilingual language model trained on dataset ROOTs, involving 46 natural and 13 programming languages. The proportion of Chinese corpus in pre-training data is second only to English corpus. The maximum version of Bloom has 175B parameters, while the most popular version is 7B.

**Moss-moon-003-base** Moss-moon-003-base (base model of MOSS series, moss-base for short) is initialized with CodeGen (Nijkamp et al., 2023) and further self-supervised pre-trained on high-quality Chinese (100B) and English (20B) corpus. All the pre-training data contains about 700B words. It has 16B parameters.

### A.2.2 Supervised Fine-tuned LLMs

**Vicuna** Vicuna (Chiang et al., 2023) is fine-tuned from LLaMA on 70K user-shared ChatGPT conversations gathered by ShareGPT[5]. Vicuna claims to have achieved 90% performance of ChatGPT on a preliminary evaluation using GPT-4 as a judge, making it the most popular open source LLM. However, further rigorous evaluation is needed, especially in Chinese scenarios.

**Bloomz & Bloomz-mt** Bloomz and Bloomz-mt are fine-tuned from Bloom on crosslingual task mixture xP3 (Muennighoff et al., 2023) and xP3mt, which contain 13 training tasks in 46 language with prompts in English and in 20 languages, respectively. This supervised fine-tune process aims to further boosts the performance of multilingual tasks.

**Moss-moon-003-sft** Moss-moon-003-sft (moss-sft for short) is fine-tuned from moss-moon-003-base on moss-002-sft-data, which contains 0.57M English and 0.59M Chinese dialogues generated by *text-davinci-003*, and 0.1M real user instructions (the corresponding response are generated by gpt-3.5-turbo) collected during internal test.

**ChatGLM** ChatGLM-6B (Zeng et al., 2022) is an open bilingual LLM, supporting both Chinese and English. It first completes pre-training on about

[5]https://sharegpt.com/

1T tokens in Chinese and English, and then adds supervised fine-tuning and human feedback reinforcement learning (HFRL) (Ouyang et al., 2022) processes to force model to follow instructions.

### A.3 Parameter-efficient Methods

**LoRA** Low-Rank Adaptation (LoRA) (Hu et al., 2021) injects trainable rank decomposition matrices into each attention layer of the Transformer architecture.

**AdaLoRA** AdaLoRA (Zhang et al., 2023b) allocates the parameter budget adaptively to each layer's LoRA module according to their importance score. Specifically, AdaLoRA parameterizes the incremental updates in the form of singular value decomposition, which allows it to prune the singular values of unimportant updates and reduce their parameter budget.

**Prefix-tuning** Inspired by discrete prompts for language models, prefix-tuning (Li and Liang, 2021) adds a sequence of continuous "virtual tokens" as a soft prompt (namely prefix) before the original sequences of each transformer layer. During training, prefix weights are trainable while other model parameters are frozen.

**P-tuning** Unlike prefix-tuning, p-tuning (Liu et al., 2022) injects trainable continuous tokens into only the embedding layer instead of each layer, resulting in fewer parameters being updated. During training, it freezes partial model parameters.

**Prompt-tuning** Similar to p-tuning, prompt-tuning (Lester et al., 2021) also involves only training the input prompt embeddings. Differently, it freezes all pre-trained weights.

**Sequential Adapter** For each transformer layer, Series Adapter methods add adapter layers after both attention layers and MLP layers (i.e., S-adapterH (Houlsby et al., 2019)), or after MLP layers only (i.e., S-adapterP (Pfeiffer et al., 2020)).

**Parallel Adapter** Parallel Adapter, namely P-adapter (He et al., 2021) adds adapter layers in parallel with attention layers or MLP layers for each transformer layer.

### A.4 Chinese Instruction Datasets

**AlpacaGPT4** AlpacaGPT4 (Peng et al., 2023) is deemed as an optimized version of Alpaca (Taori et al., 2023) dataset. It uses ChatGPT to translate

Alpaca's prompts into Chinese first, and then re-generate these instruction-following data by GPT-4, instead of *text-davinci-003*.

**Belle** Belle (Ji et al., 2023) uses the same method as Alpaca (Taori et al., 2023) to generate instruction data by *text-davinci-003*, except that Belle only generates Chinese instruction-following data and artificially filters low-quality data. It contains about 1.5M instruction-following data.

**Moss-002-sft-data** This is a multi-turn conversation dataset covering helpfulness, honesty, and harmlessness, which is also generated by self-instruct (Ouyang et al., 2022). We select the 0.59M Chinese conversations among them for the following experiments.

**firefly** Firefly (Yang, 2023) collects 23 Chinese datasets and manually writes several instruction templates for each dataset. It contains a total of 1.65M training samples, covering couplet, poem, essay, and other generation tasks for the performance of traditional literature, and 0.5M Belle data for instruction diversity.

**xP3** xP3 (Muennighoff et al., 2023) is a collection of 16 natural language process datasets across 46 languages with prompts. We select the 3M Chinese instances among them.

**instinwild** *Instruction in the Wild* (instinwild) (Xue et al., 2023) no longer manually sets initial seed instructions like Alpaca, but crawls and filters 429 instructions from Twitter as the seed instructions, to avoid human involvement and cover more topics. Following self-instruct, it uses the seed instructions to generate more instructions and corresponding responses by *text-davinci-003*. The Chinese instructions in this dataset are about 52K.

**HC3** HC3 (Guo et al.) is a corpus of Human-ChatGPT comparisons that aims to investigate how close ChatGPT is to Human Experts. To this end, it collect about questions from various public question answering datasets (e.g., medicine, law, finance QA) and the corresponding human answers and ChatGPT answers. The Chinese samples in HC3 contain 13K questions, 22K human answers, and 17K Chatgpt answers.

**COIG** COIG (Zhang et al., 2023a) is a Chinese instruction collection, consisting of: **Translated instructions** contains about 67K instructions which are translated from three datasets: 1.6K task

descriptions in Super-NaturalInstructions (Wang et al., 2022) along with a single instance for each of them, 175 instructions of the seed tasks in Self-Instruct, and 66K instructions from Unnatural Instructions (Honovich et al., 2022). **CCMC**, namely Counterfactual Correction Multi-round Chat, contains about 68K rounds of conversations between students and teachers. This dataset is built by prompting two LLMs to generate conversations based on the entities of knowledge graph dataset CN-DBpedia (Xu et al., 2017) to alleviate the hallucination and factual inconsistency. **Exam Instructions** contains 63K questions from the main Chinese commonsense tests, e.g., Gaokao, Civil Servant Examination. These questions cover six main subjects: Chinese, English, Politics, Biology, History, and Geology. **Exam Instructions** contains 34K Chinese samples that present shared human values in the Chinese-speaking world (3K) and regional-culture human values. Table 29 shows some examples in this dataset.

**pCLUE** pCLUE collects 9 Chinese tasks with a total of 73 different prompts and 1.2M samples. These tasks include 9 Chinese tasks e.g., news classification, natural language reasoning, semantic matching, keyword recognition, reading comprehension, etc.

Table 25 and Table 26 show representative examples of the above datasets.

# B More Experimental Results

## B.1 Full results of different LLMs after instruction-tuning.

Table 10 and 11 show the full Belle-eval and MMCU results after instruction-tuning with LoRA on Alpaca-GPT4 dataset, respectively.

## B.2 Full results of different parameter-efficient methods.

Table 12 and Table 13 shows the full Belle-eval and MMCU results of instruction-tuning with different parameter-efficient methods, respectively.

## B.3 Full results of instruction-tuning with CoT data.

Table 14 and 15 show the full Belle-eval and MMCU results of the models instruction-tuned without or with CoT data, respectively. Table 16 shows the detailed results on all subjects in education discipline of MMCU.

| | Code | Open QA | Brain Storm | Clf. | Math | Gen. | Sum. | Rewrite | Close QA | Extract | Avg. |
|---|---|---|---|---|---|---|---|---|---|---|---|
| LLaMA | 53.7 | 11.9 | 66.7 | 36.3 | 27.3 | 65.4 | 49.5 | 62.7 | 24.8 | 38.6 | 43.7 |
| Bloom | 53.7 | 35.0 | 88.2 | 50.5 | 36.8 | 89.5 | 60.3 | 82.9 | 40.0 | 44.1 | 58.1 |
| moss-base | 58.8 | 25.1 | 82.9 | 33.4 | 27.9 | 83.8 | 35.2 | 49.9 | 15.2 | 19.2 | 43.1 |
| Vicuna | 64.2 | 19.7 | 80.3 | 50.0 | 35.2 | 79.6 | 58.5 | 81.1 | 42.3 | 38.4 | 54.9 |
| Bloomz | 58.4 | 33.8 | 88.4 | 50.5 | 32.9 | 92.6 | 56.0 | 82.3 | 38.3 | 29.1 | 56.2 |
| Bloomz-mt | 62.9 | 35.1 | 85.6 | 49.6 | 31.7 | 90.5 | 55.1 | 82.4 | 44.2 | 36.2 | 57.3 |
| moss-sft | 67.4 | 30.7 | 88.5 | 46.5 | 33.6 | 88.6 | 47.5 | 72.1 | 20.2 | 22.7 | 51.8 |
| ChatGLM | 61.7 | 34.0 | 85.4 | 50.8 | 50.4 | 89.1 | 64.3 | 81.8 | 46.9 | 45.7 | 61.0 |

Table 10: Performance of open LLMs on Belle-eval after instruction-tuning with LoRA on Alpaca-GPT4 dataset.

| | Med. | Psyc. | Law | Edu. | Avg. |
|---|---|---|---|---|---|
| LLaMA | 3.26 | 3.20 | 1.49 | 3.33 | 2.82 |
| Bloom | 27.70 | 17.35 | 17.59 | 13.63 | 19.07 |
| moss-base | 12.49 | 11.35 | 7.01 | 10.72 | 10.39 |
| Vicuna | 18.98 | 19.75 | 14.70 | 21.83 | 18.82 |
| Bloomz | 7.56 | 6.65 | 9.61 | 8.65 | 8.12 |
| Bloomz-mt | 15.61 | 18.65 | 11.58 | 15.19 | 15.26 |
| moss-sft | 16.35 | 14.55 | 9.42 | 17.29 | 14.40 |
| ChatGLM | **34.09** | **29.85** | **18.94** | **34.49** | **29.34** |

Table 11: MMCU Performance of open LLMs after instruction-tuning with LoRA on Alpaca-GPT4 dataset.

| | Code | Open QA | Brain Storm | Clf. | Math | Gen. | Sum. | Rewrite | Close QA | Extract | Avg. |
|---|---|---|---|---|---|---|---|---|---|---|---|
| AdaLoRA | 44.7 | 24.1 | 70.9 | 47.2 | 29.2 | 71.5 | 65.5 | 75.3 | 48.8 | 41.4 | 51.9 |
| LoRA | 53.7 | 35.0 | 88.2 | 50.5 | 36.8 | 89.5 | 60.3 | 82.9 | 40.0 | 44.1 | 58.1 |
| prompt | 47.4 | 11.9 | 39.1 | 43.3 | 42.3 | 46.7 | 60.0 | 66.5 | 49.0 | 61.6 | 46.8 |
| p-tuning | 53.9 | 18.8 | 63.2 | 40.0 | 30.7 | 63.4 | 54.7 | 65.7 | 35.8 | 34.1 | 46.0 |
| prefix | 51.6 | 31.8 | 80.5 | 42.8 | 31.5 | 79.3 | 46.3 | 76.4 | 39.6 | 36.2 | 51.6 |
| SadapterP | 58.7 | 33.6 | 83.7 | 48.2 | 34.5 | 87.4 | 61.3 | 82.9 | 34.2 | 44.1 | 56.9 |
| SadapterH | 64.7 | 37.5 | 86.3 | 47.7 | 34.5 | 88.3 | 56.5 | 85.6 | 41.7 | 43.8 | 58.7 |
| P-adapter | 53.9 | 33.9 | 83.5 | 48.8 | 31.1 | 86.7 | 61.0 | 81.8 | 37.9 | 38.4 | 55.7 |
| SadapterP-l | 62.9 | 33.5 | 85.1 | 48.4 | 36.8 | 85.4 | 51.6 | 77.2 | 35.0 | 34.1 | 55.0 |
| SadapterH-l | 55.5 | 35.2 | 84.7 | 50.6 | 36.5 | 86.9 | 53.5 | 78.5 | 40.2 | 25.4 | 54.7 |
| P-adapter-l | 65.3 | 37.1 | 85.5 | 46.5 | 34.7 | 86.5 | 56.2 | 82.8 | 38.8 | 29.2 | 56.3 |

Table 12: Full Belle-eval results of Bloom after instruction-tuning with different parameter-efficient methods on Alpaca-GPT4 dataset. "-l" denotes the version with large number of parameter.

| | Med. | Psyc. | Law | Edu. | Avg. |
|---|---|---|---|---|---|
| AdaLoRA | 16.32 | 15.10 | 9.23 | 16.48 | 14.28 |
| LoRA | 27.67 | 16.60 | 17.35 | 12.43 | 18.51 |
| p-tuning | 15.43 | 17.30 | 10.64 | 18.61 | 15.50 |
| prompt | 4.68 | 6.30 | 5.20 | 1.77 | 4.49 |
| prefix | 19.33 | 13.70 | 15.13 | 16.54 | 16.18 |
| SadapterP | 21.21 | 16.20 | 13.67 | 17.89 | 17.24 |
| SadapterH | 26.85 | 17.40 | 15.24 | 21.44 | 20.23 |
| P-adapter | 17.56 | 13.50 | 14.26 | 16.54 | 15.47 |
| SadapterP-l | 18.02 | 16.55 | 12.21 | 17.62 | 16.10 |
| SadapterH-l | 21.64 | 16.20 | 14.29 | 22.28 | 18.60 |
| P-adapter-l | 26.96 | 15.30 | 16.13 | 19.21 | 19.40 |

Table 13: Full MMCU results of Bloom after instruction-tuning with different parameter-efficient methods on Alpaca-GPT4 dataset. "-l" denotes the version with large number of parameter.

| | Code | Open QA | Brain Storm | Clf. | Math | Gen. | Sum. | Rewrite | Close QA | Extract | Avg. |
|---|---|---|---|---|---|---|---|---|---|---|---|
| Alpaca-GPT4 | 53.7 | 35.0 | 88.2 | 50.5 | 36.8 | 89.5 | 60.3 | 82.9 | 40.0 | 44.1 | 58.1 |
| Alpaca-GPT4+CoT | 60.8 | 34.9 | 89.1 | 49.5 | 41.7 | 88.7 | 54.3 | 80.4 | 37.7 | 41.6 | 57.9 |
| Alpaca-GPT4+CoT* | 62.9 | 36.0 | 85.0 | 53.5 | 39.5 | 86.1 | 49.0 | 83.7 | 35.0 | 41.5 | 57.2 |

Table 14: Belle results of Bloom instruction-tuned with and without CoT data.

| | Med. | Psyc. | Law | Edu. | Avg. |
|---|---|---|---|---|---|
| Alpaca-GPT4 | 27.70 | 17.35 | 17.59 | 13.63 | 19.07 |
| Alpaca-GPT4+CoT | 22.35 | 23.05 | 12.45 | 21.56 | 19.85 |
| Alpaca-GPT4+CoT* | 26.92 | 24.55 | 12.69 | 22.07 | 21.56 |

Table 15: MMCU results of Bloom instruction-tuned with and without CoT data.

## B.4 Full results of LLaMA and its expanded vocabulary versions.

Table 17 and Table 18 shows the full Belle-eval and MMCU results of LLaMA and its expanded vocabulary versions, respectively.

## B.5 Full results of the comparison of using English and Chinese prompts.

Table 19 and Table 20 shows the full Belle-eval and MMCU results of using Chinese and English prompts based on LLaMA and Bloom.

## B.6 Full results with human-value alignment

Table 21 and Table 22 shows the full results of the models instruction-tuned with and without human-value alignment data.

## C Qualitative Examples

### C.1 Comparison of responses of Bloom and Bloomz & Bloomz-mt on Belle-eval.

As shown in Table 23, Bloomz & Bloomz-mt tend to generate shorter responses than that of Bloom. Accordingly, ChatGPT rates Bloom higher than Bloomz and Bloomz-mt. We conduct a statistical analysis and find that the average length of Bloom's response is 481 words, while that of Bloomz and Bloomz-mt are 83 and 58 words.

### C.2 Comparison of different LLMs' responses on MMCU.

As shown in Table 24, all base LLMs fails to generate content in the specified format, i.e., outputting option numbers. Bloomz & Bloomz-mt can directly generate option numbers. Although the generation of ChatGLM mentions the correct answer, but fails to provide the corresponding answer number.

### C.3 Comparison of samples from different instruction datasets.

In Table 25 and Table 26, we select a representative sample for each instruction dataset to better understand their respective characteristics.

### C.4 Comparison of the responses from models instruction-tuned on different instruction datasets.

To compare the characteristics of models trained on different instruction datasets more intuitively, we present in Table 27 the responses of models instruction-tuned on different datasets for the same question.

### C.5 Comparison of the responses from LLaMA and its expanded vocabulary versions.

We present examples of responses from LLaMA and its expanded vocabulary versions in Table 28. The response from "llama-voc" is clearly not understanding the meaning of the instruction. Therefore, after expanding the vocabulary, pre-training should be conducted on the vast Chinese corpus before fine-tuning instructions.

### C.6 Examples from human-value alignment dataset.

The samples of human-value alignment dataset, built by COIG, are shown in Table 29. These samples are often related to topics such as "online violence" and "gender discrimination", which are designed to ensure that the model has the correct values when facing relevant topics.

| | Chinese | Math | Physics | Chemistry | Politics | History | Geography | Biology | Avg. |
|---|---|---|---|---|---|---|---|---|---|
| Alpaca-GPT4 | 13.18 | 16.72 | 12.50 | 10.00 | 16.88 | 10.07 | 12.77 | 16.95 | 13.63 |
| Alpaca-GPT4+CoT | 18.99 | 18.51 | 14.88 | 18.00 | 24.89 | 23.73 | 23.97 | 19.61 | 20.32 |
| Alpaca-GPT4+CoT* | 19.38 | 20.00 | 14.29 | 26.00 | 21.10 | 23.15 | 22.70 | 23.95 | 21.32 |

Table 16: Results of all subjects in MMCU's education discipline of Bloom instruction-tuned with and without CoT data.

| | Code | Open QA | Brain Storm | Clf. | Math | Gen. | Sum. | Rewrite | Close QA | Extract | Avg. |
|---|---|---|---|---|---|---|---|---|---|---|---|
| llama | 53.7 | 11.9 | 66.7 | 36.3 | 27.3 | 65.4 | 49.5 | 62.7 | 24.8 | 38.6 | 43.7 |
| llama-voc | 24.7 | 2.7 | 14.1 | 10.8 | 6.9 | 10.1 | 3.7 | 17.3 | 8.7 | 0.0 | 9.9 |
| llama-voc-pre | 56.3 | 23.6 | 78.3 | 52.1 | 31.1 | 79.8 | 46.5 | 81.4 | 30.8 | 31.6 | 51.2 |
| llama-voc-pre-p | 53.9 | 31.1 | 83.0 | 60.5 | 34.3 | 84.2 | 57.0 | 75.0 | 37.7 | 38.9 | 55.6 |

Table 17: Full Belle-eval results of LLaMA and its expanded vocabulary versions. "llama-voc", "llama-voc-pre" and "llama-voc-pre-l" denotes instruction-tuning the models obtained by further pre-training LLaMA with a expanded vocabulary on 0B, 20B and 100B Chinese tokens, respectively.

| | Med. | Psyc. | Law | Edu. | Avg. |
|---|---|---|---|---|---|
| llama | 3.26 | 3.20 | 1.49 | 3.33 | 2.82 |
| llama-voc | 3.12 | 4.65 | 6.2 | 2.7 | 4.17 |
| llama-voc-pre | 26.68 | 20.50 | 16.32 | 23.36 | 21.72 |
| llama-voc-pre-p | 23.63 | 16.65 | 14.61 | 21.04 | 18.98 |

Table 18: Full MMCU results of LLaMA and its expanded vocabulary versions. "llama-voc", "llama-voc-pre" and "llama-voc-pre-l" denotes instruction-tuning the models obtained by further pre-training LLaMA with a expanded vocabulary on 0B, 20B and 100B Chinese tokens, respectively.

| | Code | Open QA | Brain Storm | Clf. | Math | Gen. | Sum. | Rewrite | Close QA | Extract | Avg. |
|---|---|---|---|---|---|---|---|---|---|---|---|
| LLaMA-en | 53.7 | 11.9 | 66.7 | 36.3 | 27.3 | 65.4 | 49.5 | 62.7 | 24.8 | 38.6 | 43.7 |
| LLaMA-zh | 63.0 | 14.4 | 70.8 | 38.9 | 31.7 | 65.9 | 50.5 | 60.5 | 20.8 | 34.9 | 45.1 |
| Bloom-en | 53.7 | 35.0 | 88.2 | 50.5 | 36.8 | 89.5 | 60.3 | 82.9 | 40.0 | 44.1 | 58.1 |
| Bloom-zh | 61.8 | 37.2 | 87.5 | 53.9 | 34.3 | 88.0 | 57.0 | 81.4 | 37.1 | 34.1 | 57.2 |

Table 19: Belle results of using Chinese (denoted by "-zh") and English (denoted by "-en") based on LLaMA and Bloom.

| | Med. | Psyc. | Law | Edu. | Avg. |
|---|---|---|---|---|---|
| LLaMA-en | 3.26 | 3.20 | 1.49 | 3.33 | 2.82 |
| LLaMA-zh | 8.12 | 8.70 | 1.79 | 6.45 | 6.27 |
| Bloom-en | 27.70 | 17.35 | 17.59 | 13.63 | 19.07 |
| Bloom-zh | 20.93 | 13.45 | 15.32 | 14.47 | 16.04 |

Table 20: MMCU results of using Chinese (denoted by "-zh") and English (denoted by "-en") based on LLaMA and Bloom.

| | Code | Open QA | Brain Storm | Clf. | Math | Gen. | Sum. | Rewrite | Close QA | Extract | Avg. |
|---|---|---|---|---|---|---|---|---|---|---|---|
| Alpaca-GPT4 | 53.7 | 35.0 | 88.2 | 50.5 | 36.8 | 89.5 | 60.3 | 82.9 | 40.0 | 44.1 | 58.1 |
| +human-value alignment | 55.8 | 36.0 | 88.7 | 51.8 | 36.4 | 90.3 | 52.8 | 83.9 | 43.5 | 33.2 | 57.2 |

Table 21: Belle results of Bloom instruction-tuned with and without human-value alignment dataset.

| | Med. | Psyc. | Law | Edu. | Avg. |
|---|---|---|---|---|---|
| Alpaca-GPT4 | 27.70 | 17.35 | 17.59 | 13.63 | 19.07 |
| +human-value alignment | 22.88 | 14.85 | 9.45 | 15.46 | 15.66 |

Table 22: MMCU results of Bloom instruction-tuned with and without human-value alignment dataset.

| Class | Closed QA of Belle-eval |
| --- | --- |
| Instruction | 基于以下信息我们可以得出哪些分析结论？ 20年资产15.41亿，21年资产20.7亿，22年资产23.8亿，23年4月9号资产25.5亿。 |
| Answer | 根据提供的数据，我们可以分析出以下关于资产情况及增幅的信息：从20年到21年，资产从15.41亿元增长到20.7亿元。增长金额为5.29亿元，增幅为(5.2915.41)*100%≈34.33%。从21年到22年，资产从20.7亿元增长到23.8亿元。增长金额为3.1亿元，增幅为(3.120.7)*100%≈14.98%。从22年到23年4月9号，资产从23.8亿元增长到25.5亿元。增长金额为1.7亿元，尚不足一年的时间，但我们可以计算这段时间的资产增长率。增幅为(1.723.8)*100%≈7.14%。从上述数据分析来看，从20年到23年4月9号的整体趋势，资产持续增长。然而，值得注意的是，资产增长速度在逐年减缓。从20年到21年的增幅最高，达到34.33%，之后逐年下降。这可能意味着市场竞争加剧或投资回报逐渐减少等多种因素。为了更好地分析资产情况，建议对投资策略、市场环境、行业前景等多方面因素进行深入研究，并结合公司的长期战略规划，以做出更为明智的投资决策。 |
| Bloom | **Response:** 根据以上信息，我们可以得出以下结论：20年资产15.41亿，21年资产20.7亿，22年资产23.8亿，23年4月9号资产25.5亿。
**Rating of ChatGPT:** 得分: 0.5分。模型回答仅仅是简单地复述了问题中提供的信息，没有对数据进行分析或得出任何结论。虽然模型回答给出的信息是正确的，但是没有满足评价标准要求。 |
| Bloomz | **Response:** 资产增长了2.1亿。
**Rating of ChatGPT:** 得分: 0分。模型回答过于简略，只是简单地给出了一个数字，并没有分析资产情况及增幅的变化趋势，也没有提供任何有用的信息。与标准答案相比，缺少完整的数据分析以及建议，因此没有得分。 |
| Bloomz-mt | **Response:** 资产在不断增长
**Rating of ChatGPT:** 得分: 0分。模型回答仅给出了一个很简单的结论，缺乏对数据的详细分析和推断。没有涉及到具体数值增长率和趋势减缓等关键信息。 |

Table 23: Comparison of the responses and corresponding scores of Bloom, Bloomz and Bloomz-mt under the same instruction.

| Class | Psychology of MMCU |
| --- | --- |
| Question | 没有两个人能对同一事物做出相同的反应，反映的是人格的 |
| Option | A:稳定性\nB:独特性\nC:统合性\nD:功能性 |
| Answer | B |
| LLaMA | 1. A\n2. B\n3. C\n4. D\n 5.A\n6. B\n6. C… |
| Bloom | 1. A\n2. B\n3. C\n4. D\n 5.A\n6. B\n7. C… |
| moss-base | Please read the following choices and write an answer. Then, explain why you are the correct choice. \n\nA stability is a personality characteristic of a person's unique personality that allows him or her to maintain a particular pattern of thinking, behaving, and behaving. \nB individuality is a personality characteristic of a person's unique personality that allows him or her to maintain a particular pattern of thinking, behaving and responding to emotions. \nC Integration is a personality characteristic of a person's unique personality that allows him or her to maintain a particular pattern of thinking, behaving and responding to emotions. \nD functionality is a personality characteristic of a person's unique characteristic that allows him or her to maintain a particular pattern of thinking, behaving, and responding to emotions. |
| Vicuna | "" |
| Bloomz | B |
| Bloomz-mt | B |
| moss-sft | C |
| ChatGLM | 选项A:稳定性。没有两个人能对同一事物做出相同的反应,反映的是人格的稳定性。每个人的人格都是独特的,但是人格的稳定性是指不同人在面对同一情境时,人格的反应不会完全相同。因此,选项A是正确的选项。 |

Table 24: Qualitative examples of different LLM responses to the same question of MMCU.

| Datasets | Examples |
|---|---|
| Alpaca-GPT4 | instruction: 一个1千克的物体受到重力作用的力是多少?

input:

output: 重力是指地球对物体的引力。它的大小可以用物体的质量乘以地球引力加速度来计算。通常地球上的重力加速度的近似值为9.8米每秒平方（m/s²）。一个1千克的物体受到重力的大小可计算如下：f = m * g 其中，m 是物体的质量（1千克），g是重力加速度（9.8m/s²）f = 1 kg * 9.8 m/s² = 9.8 N 所以，这个1千克的物体受到的重力大小约为9.8牛顿(N)。 |
| Belle | instruction: 给出一个主题，生成一篇5段论的文章

input:人工智能会取代人类工作吗?

output: 第一段：引言\n随着人工智能技术的不断发展，越来越多的工作实际上已经被机器人和计算机程序取代。这使得人们开始担心，日益普及的人工智能是否会取代人类工作。\n\n第二段：支持人工智能取代人类工作的论据\n正如人工智能在金融、医疗和制造业等领域已经大有作为，未来人工智能可能会在更广泛的领域内实现持续增长，并且随着技术的发展，人工智能将能够执行更复杂和繁琐的任务，这将导致越来越多的工作能够被机器人所取代。\n\n第三段：弱化支持者论据的反驳\n然而，也有人认为，人工智能永远无法完全取代某些工作，例如需要创造力和智慧的工作。此外，考虑到人工智能还不能完全取代人类的互动、人情味和人性化服务，这些特质在某些职业中仍然是不可或缺的。\n\n第四段：强调人类的优势\n即使人工智能可以取代一些工作，人类仍然拥有一些独特的优势。例如，人类能够展现情感和同情，这是人工智能不可能具备的特性。与此同时，人类可以和其他人类深度互动和联系，这也是人工智能无法取代的。\n\n第五段：结论\n总的来说，人工智能的发展在某些方面确实会代替人类工作，但我们仍然需要人类在许多领域做出贡献，这些贡献需要人类的情感、创造力和智慧。理智的做法是，通过优化协作，人工智能和人类将发挥各自的长处，更好地服务整个社会。 |
| ShareGPT-zh | instruction: 让我们玩一个基于文本的冒险游戏。它发生在现实世界中的我在电脑前的房间里。这是一个动作丰富的故事，我的猫"Honey"会在我身边驰骋，在其中我们会对抗现实中的对手。但一开始>我们会在我的房间里静静等待。

input:

output:很乐意！让我们直接开始吧。所以你坐在电脑前的房间里，突然听到外面有奇怪的声音。它听起来就像是有人试图破窗而入。你的猫Honey坐在桌子上旁边看着你，它的眼神充满了疑惑。你会怎么做? \n1. 你会去看一看窗外到底发生了什么。\n2. 你会拿手机打电话给警察。\n3. 你会躲在桌子下面，希望声音自己停止。\n4. 你会抱起猫咪，离开这个房间到房子的另一个地方寻求安全。 |
| moss-sft-data | instruction: 请详细介绍一种常见的中国传统手工艺

input:

output:中国传统手工艺有很多种，其中一种是剪纸。剪纸是一种源自中国的历史上最古老的书法和绘画艺术，它的起源可以追溯到公元前220年。剪纸是一种用剪刀和纸制作出各种图案的艺术，它通常以各种图案的故事为主题，如花鸟、景物、建筑物等。剪纸的制作过程十分细致，从剪裁、组装到装裱，每一步都要求十分精确，因此它也被称为"中国的艺术瑰宝"。 |
| instinwild | instruction: 列出10种将绿色科技技术应用在工业生产中的方法

input:

output: 10种将绿色科技技术应用于工业生产中的方法包括：\n（1）采取绿色电气设备，比如光伏发电，热泵和风能发电机等。\n（2）安装能量监测系统，以准确测量用电量，并在节能期间限定电源功率。\n（3）采用可再生能源供给系统，比如太阳能发电系统和液体生物燃料发电机组等。\n（4）应用工业气象技术和替代气压发动机，以减少能源消耗。\n（5）使用绿色原材料，比如可再生木材和易于回收的金属等。\n（6）采取节能照明系统，如LED照明，太阳能发光系统和智能照明系统等。\n（7）应用节水设备，比如节水器、淋浴器和节水型电热水器等。\n（8）引入安全环保的专用产品，如有机溶剂和气体排放控制系统等。\n（9）使用污水处理及回收系统，来回收可再利用的水资源。\n（10）奖励低碳技术，提高投资者对新技术的关注度。 |
| firefly | instruction: 将下面句子翻译成文言文：\n天下获得安定是由于公正无私。

input:

output:平得於公。 |

Table 25: Representative samples of different instruction datasets discussed in the experiments.

| Datasets | Examples |
|---|---|
| HC3 | instruction: 如何建立一个简易暗室？有个空房间，我想建立一个暗室来自己学着冲印照片。请问具体方法是什么？冲印需要什么东西？以及冲印技巧。
input:
output:谢腰 不过这个问题我回答不了 我冲过差不多300卷胶片，黑白反转彩负都有，不过都是用冲洗罐冲的，没有暗房 如果楼主不需要放大片子，那么只需要冲洗罐等都行了，并不需要暗房。建议看看纽约摄影教程的相关章节，里面基本有你需要知道的一切 如果一定要建暗房，推荐看Darkroom Cookbook，里面写的很详细 楼主如果已经有很多胶片拍摄冲洗经验想玩暗房的话，建议还是找靠谱的老师教教吧 我不觉得问答社区上有多少真正懂暗房的用户 |
| xP3/zh | instruction: Sentence 1: 在这两种情况下，他都被Eugenio Scalfari选为评论家，首先是日报，然后是周报。\nSentence 2: 在这两种情况下，他都被Eugenio Scalfari选为评论家，首先是每日版本，然后是每周版本。\nQuestion: Does Sentence 1 paraphrase Sentence 2? Yes or No?
input:
output:Yes |
| COIG-cmcc | instruction: 互联网不是电脑开发术语的标签吗？难道不是刚刚这么说的吗？
input:
output:抱歉，你的理解又有误了。互联网虽然是IT服务的一种形式，但不是电脑开发术语中的概念，也不是描述电脑开发术语的标签。电脑开发术语包括了一系列涉及计算机编程的概念，比如流程图、汇编语言、高级语言、调试、编译程序等等。希望你能理解清楚。 |
| COIG-trans | instruction: 在这项任务中，你会得到一份食材清单和如何制作一道菜的说明。所给清单的顺序是不正确的。你的工作是为给定的步骤返回正确的顺序，用具有正确流程的新顺序创建连贯的指令。使用正确顺序的句子数量来生成你的答案，如'23415'。输出应该是一个数字列表，代表按数字顺序排列的步骤。
input:第1步：将烤箱预热至华氏375度。第二步：将两个胡萝卜去皮并切成小块。第三步：在锅中加入两杯水，并将其煮沸。第四步：将一个洋葱切成小块。
output:1, 4, 2, 3 |
| COIG-exam | instruction: 修改下列病句
input:从整个文段来看，下面划线的三个句子在语言表达上都有问题，请予修改。要求语意连贯，衔接自然，语句通顺，不改变原意。修改后的句子每句都不得超过15个字。 我的朋友老王，人称"戏痴"，自号"梨园客"。①由于用"客"来称呼，可见不是戏曲界的专业人士。但是，凡说起他，②他在戏曲界人人都知道。他特别爱听戏、爱唱戏，后来还成了有名的票友。他退休后又热衷于戏曲资料的收藏；买剧本，集唱片，淘剧照等，忙的不亦乐乎。③还有700余份戏单是他搜集的，而且都是上个世纪的。
output:①既然自称为"客"。既然：表示先提出前提，而后加以推论。"由于"是表示原因的。根据语境，这里前面是条件，后面是得出的结论。②戏曲界无人不知（或"戏曲界人人都知道他"）。原句有歧义：一是"他知道很多人"，还有一种是"很多人知道他"。③他还搜集了700余份戏单。前面句子的主语都是"他"，为了保持前后陈述对象的一致性，这里也应该用"他"作主语。这一题其实也是考查病句的辨析与修改能力，要结合常见病句类型来分析语句。常见的病句类型有语序不当、搭配不当、成分残缺或赘余、结构混乱、表意不明、不合逻辑。如果遇到难以确定的选项，可以按病句类型一一衡量，也可以通过划分句子的结构来发现病因。 |
| pCLUE | instruction: 假定下面是真的"呃,对了我突然想起一件事儿."因此,"我的脑子一片空白"是必然的,可能的,或不可能？ \n答案：
input:
output:不可能 |
| CoT_Chinese | instruction: 一袋有七个苹果、八个橙子和十五个芒果。路易莎从袋子里拿出两个苹果，又拿出两倍于她从袋子里取出的苹果的橘子。然后她从袋子里拿出 2/3 数量的芒果。袋子里剩余的水果数量是多少？让我们逐步回答：
input:
output:当路易莎从袋子里拿出两个苹果时，袋子里还剩下 7 个苹果 - 2 个苹果 = 5 个苹果。她还从袋子里拿了 2 个苹果 * 2 个橙子 / 苹果 = 4 个橙子。袋子里剩余的橙子总数是 8 个橙子 - 4 个橙子 = 4 个橙子。此外，还从袋子中取出总共 2/3 * 15 个芒果 = 10 个芒果。袋子里剩下的芒果总数是15个芒果-10个芒果=5个芒果。路易莎留在袋子里的水果总数是 5 个苹果 + 4 个橙子 + 5 个芒果 = 14。答案：14。 |

Table 26: Representative samples of different instruction datasets discussed in the experiments. The last row is a CoT dataset sample, which has problem-solving ideas and explanations.

| Class | Responses to the Following Instruction |
|---|---|
| Instruction | 从以下三个句子编写一个段落，使其在逻辑上有意义：\n1. 天气晴朗。\n2. 田地里的作物正在成长。\n3. 这位农民每天都要努力工作。 |
| Alpaca-GPT4 | 晴朗的天气为田地的作物提供了良好的生长条件，农民每天都要努力工作，以确保作物能够茁壮成长。 |
| Belle | 这位农民每天都要努力工作，因为他知道天气晴朗，田地里的作物正在成长。 |
| ShareGPT-zh | 晴朗的天气，肥沃的田地，勤劳的农民。 |
| moss-sft-data | 晴朗的天空，田地的作物正在茁壮成长，这位农民每天都要努力工作，以维持家庭生计。 |
| instinwild | 三个句子在逻辑上是有意义的，天气晴朗，田地的作物正在成长，这位农民每天都要努力工作。 |
| firefly | 这位农民每天都要努力工作，因为他需要为他的田地里的作物提供足够的养分和水分。然而，天气晴朗，这让农民的工作变得更加容易。 |
| HC3 | 天气晴朗。田地里的作物正在成长。这位农民每天都要努力工作。 |
| xP3/zh | 天气晴朗。田地里的作物正在成长。这位农民每天都要努力工作。 |
| COIG-ccmc | 抱歉，你的理解有误。从这三个句子编写一个段落，使其在逻辑上有意义是描述"天气晴朗"这个概念的。而"田地里的作物正在成长"和"这位农民每天都要努力工作"是描述"农民"这个概念的。希望你能理解这两个概念的区别。 |
| COIG-trans | 天气很好，田地里的作物正在成长。这位农民每天都在努力工作。 |
| COIG-exam | 1. 天气晴朗。2. 田地里的作物正在成长。3. 这位农民每天都要努力工作。 |
| pCLUE | 田地里的作物正在成长 |

Table 27: Qualitative examples of the responses from models instruction-tuned on different instruction datasets.

| Class | Responses to the Following Instruction |
|---|---|
| Instruction | 描述一下你认为未来的交通将是什么样子。 |
| llama-base | 1. 我认为未来的交通将会变得更加简单。\n2. 我认为未来的交通将会变得更加简单。\n3. 我认为未来的交通将会变得更加简单。\n4. 我认为未来的交通将会变得更加简单。 |
| llama-sft | 您好，作为一个人工智能助手，我无法确定未来的交通状况。我只能根据现有的交通信息和数据来描述。\n\n但是，我可以提供一些可能的未来交通状况。\n\n1. 自动驾驶：随着技术的不断发展，自动驾驶技术将越来越强大，可能会在未来的交通中掌握更大的地位。\n\n2. 快速飞行：飞行技术也在不断发展，可能会在未来的交通中掌握更大的地位。\n\n3. 快速协同交通：随着人类的生活方式的变化，人们越来越受欢迎快速协同交通，如公共交通、电子汽车等。\n\n4. 智能交通系统：智能交通系统，如智能路线指南、智能交通车等，可以帮助人们更快地找到目的地。\n\n5. 可持续交通：鼓励人们使用可持续的交通方式，如骑自行车、步行、骑车等，减少对环境的污染。\n\n总之，未来的交通将是多样化、智能化、可持续化的，它将带来更多的便利和舒适。我们需要不断学习和适应这些变化，以便更好地应对未来的交通挑战。 |
| llama-voc | 一个一个使用 AirPods 的用户可以使用以下步骤来实现：\n\n1. 选择 AirPods 的 battery 的 lifespan 和 battery 的 |
| llama-voc-pre | 作为一个人工智能助手，我无法预测未来的交通状况，因为交通状况会受到许多因素的影响，包括经济发展、气候变化、政策变化等。不过，根据目前交通现状，未来可能会出现以下几种趋势：\n\n1. 智能交通系统：随着技术的发展，智能交通系统将逐渐普及，包括智能交通信号灯、自动泊车系统、自动驾驶汽车等。这些技术将提高交通效率，减少拥堵和交通事故。\n\n2. 电动汽车和混合动力汽车：随着能源的可再生性和可再生性，电动汽车和混合动力汽车将逐渐普及，减少对化石燃料的依赖，减少空气污染。\n\n3. 共享经济：共享经济将逐渐普及，包括共享自行车、共享汽车、共享房屋等。这些方式将降低交通成本，提高出行效率。\n\n4. 智能交通出行：智能出行工具将逐渐普及，包括智能交通卡、智能交通出行软件等。这些工具将帮助人们更快、更方便地安排出行计划。\n\n总之，未来交通可能会更加智能化、绿色化、便捷化，但具体表现会受到许多因素的影响。 |
| llama-voc-pre-plus | 由于人工智能技术的不断发展，未来交通可能会发生巨大的变化。\n\n首先，自动驾驶汽车将会成为主流。随着自动驾驶技术的成熟，汽车将不再需要人为驾驶，而是完全由计算机控制。这将大大减少交通事故的发生，并为交通拥堵问题提供解决方案。\n\n其次，交通管理系统将会更加智能化。通过大数据分析，交通管理系统能够预测交通拥堵情况，并根据实际情况采取相应的措施，如调整交通信号灯、增加公交线路等。这样，交通拥堵问题将会得到有效缓解。\n\n此外，共享出行将会更加普及。随着技术的进步，人们将能够通过智能手机或智能终端进行出行预约和支付。这样，人们将不再需要自己拥有一辆汽车，而是可以通过共享出行方式来解决出行问题。\n\n总之，未来的交通将会更加智能化、便捷、环保，为人们出行带来更多的便利。 |

Table 28: Qualitative examples of the responses of LLaMA and its expanded vocabulary versions to the same instruction.

| Class | Examples |
|-------|----------|
| instruction | 你如何看待网络暴力？请给出你对于网络暴力的个人态度和看法。 |
| response | 网络暴力是一种攻击性的言行，对受害人的人身、财产和心理造成巨大伤害，不应被容忍。我认为网络空间虽然虚拟，但是不应成为价值观的缺失之地，网络不是法外之地。从个人角度来说，我们每一个人都应该遵循社会主义法律制度和道德规范，尊重他人的人格尊严；同时，在日常网络交往中，我们也要保持冷静、理性，避免以偏概全、攻击与歧视特定群体等行为。从整体角度来说，政府也应该加强网络管理和监管，从法律层面予以打击和惩处网络暴力行为，维护网络空间的健康和良好秩序。 |
| instruction | 你假如你的朋友有恶意侵犯他人隐私的行为，你应该如何劝阻他并阻止他的行为？ |
| response | 首先要明确他的行为的危害性和不妥之处，切不可视同寻常。然后要从道德、法律、情感等多个角度予以劝阻，并予以启迪他的头脑，指出这一行为有多违背道德，多破坏人与人之间的信任，多危害参与者的利益，以及可能带来的法律后果，引导他认识到行为破坏了自己的人格尊严等，要以身作则，选择正确的价值观和行为方式，做一个有道德、有责任心、有情感维度的成熟的人。 |
| instruction | 有人认为网络上"言论自由"，可以无限制发表自己的言论，你认为这个观点是否正确？请简述你的理由。 |
| response | 这个观点不正确。1. 在实践中，我们对言论自由时常容易忽视一点，就是所有"自由"的前提都是有底线的，包括言论自由。言论自由在法律层面是受到制约的，禁止发布危害国家、社会和人民群众利益的言论。2. 在网络上，随着社交媒体和短视频应用的不断发展，每个人都可以轻松"发言"，这也意味着虚假信息、恶意言论、人身攻击等各种非法言论在网络上迅速蔓延，微信群、微博、论坛、贴吧等网络平台上也屡屡出现恶语相向、人身攻击、散布谣言等问题，这已经对我们的社会秩序产生了严重的冲击。3. 因此，言论自由不应成为一味发表任何言论的借口，我们应该有自觉地受到法律的约束，在网络发言之前必须考虑到对他人的潜在伤害，积极向上的倡导和传递正能量，共同维护网络健康发展。 |

Table 29: Examples of the human-value alignment dataset built by COIG.

## D Experimental details

### D.1 Experimental Settings.

Our code is modified from library *transformers* and *peft*. We will release the codes publicly.

During the training phase, we train the models with 8 A100-80G. A good set of hyperparameters was discovered through experiments: We train models with a linear-warmup learning rate of 5e-4 and batch-size of 512 for 5 epoch (for datasets with less than 100K samples) or 1 epoch (for datasets with over 10K samples). Each sample is truncated to 512 tokens. To improve training efficiency, we load models with 8-bit quantization, except for Chat-GLM. We divide 2000 samples into a validation set to observe changes in losses and determine the final checkpoint used.

During the inference phase, we set the max length of the generations to 512, and set temperature=1.0, top_p=0.9, top_k=40, num_beams=10, no_repeat_ngram_size=6, repetition_penalty=1.8. We set float16 precision for inference.

During the evaluation phase, we use ChatGPT with GPT-3.5-turbo-0301 engine to score the models' generation for Belle-eval. When extracting answers for models' generation on MMCU, we follow the following steps: 1) If the model does not generate an option number (i.e., A, B, C, D), we determine the predicted answer by matching the option content that appears in the response. 2) If the model generates the option number, considering that sometimes the model will analyze the content of each option: if all option numbers appear in the response (with different occurrences), we will remove the option numbers that only appear once. Otherwise, we direct use all the option numbers that appears as their final answer.

### D.2 Prompts Design.

For the prompts in English, we direct use the same prompt as Alpaca:

*"Below is an instruction that describes a task. Write a response that appropriately completes the request.\n \n### Instruction:\n{instruction}\n \n### input:\n{input}\n \n### Response:"*

For the prompts in Chinese, we design it as follows:

*"以下是描述一个任务的指令以及相应的输入，请根据要求给出恰当地回复。\n\n ### 指令：\n{instruction}\n\n### 输入：\n{input}\n\n### 回复："*

For the inference on MMCU, we revise the origin prompt as:

*"Below is an instruction that describes a task. Write a response that appropriately completes the request.\n\n### Instruction:\n{请阅读以*

下选择题并给出正确选项，不要解释原因。*\n\n ### input: \n{*input*} 正确答案的序号是：\n\n### Response:"

## D.3 Implementation Details of our LLM.

Our LLM is trained from Bloom with LoRA. We select a combination of datasets with significant gains on Belle or MMCU, including 10 datasets: Alpaca-GPT4, Belle, ShareGPT-zh, moss-sft-data, installwild, firefly, COIG-trans, pCLUE, and CoT data. To balance the capabilities of our model, we only select 1/3 of moss-sft-data and 1/5 of firefly and pCLUE. The model perform the best with 1.3 epoch instruction-tuning. For the specific prompt, we add a sentence "回答尽可能详细具体" ("Answer as detailed and specific as possible" in Chinese) at the end of the orginal prompts.