# OpenReview forum: "An Empirical Study of Instruction-tuning Large Language Models in Chinese"
_EMNLP/2023/Conference — EMNLP 2023 Findings_

### Official Review · Reviewer_CCaJ · 2023-07-26

**Soundness:** 4

**Excitement:**

3: Ambivalent: It has merits (e.g., it reports state-of-the-art results, the idea is nice), but there are key weaknesses (e.g., it describes incremental work), and it can significantly benefit from another round of revision. However, I won't object to accepting it if my co-reviewers champion it.

**Paper Topic And Main Contributions:**

The present research deals with the use and adaptation of LLMs for Chinese language. Different LLMs are evaluated, as well as methods for improvement.

The evaluation is based on two evaluation benchmarks with NLP tasks, but also programming and mathematics tasks. The performance of different LLMs with Chinese language is compared on these benchmarks.

Especially the LLM "Bloom" can benefit from "instruction tuning". The LoRA method leads to the best results. Instruction datasets with Chinese instructions are used for training. Other factors that are investigated: Vocabulary (especially in the tokenizer), language of the prompt (English or Chinese), human values to avoid generating toxic content.

The resulting LLM will be made publicly available.


**Reasons To Accept:**

A new LLM has been created and will be made publicly available. The research here is on the Chinese language.

The paper is well-written and understandable.

**Reasons To Reject:**

I cannot think of any.

**Reproducibility:**

4: Could mostly reproduce the results, but there may be some variation because of sample variance or minor variations in their interpretation of the protocol or method.

**Reviewer Confidence:**

4: Quite sure. I tried to check the important points carefully. It's unlikely, though conceivable, that I missed something that should affect my ratings.

**Typos Grammar Style And Presentation Improvements:**

In the bibliography, it is not useful to write all names of all contributors, if it is a very long list.

---

> ### Author Rebuttal · Authors · 2023-08-26
>
> + Thanks for the agreement and constructive comments.
> + Given that you did not provide any reasons for rejecting this paper, but only scored 3 points for "excitement",  we respectfully speculate that you are concerned about our "novelty" . However, we call for an inclusive mindset towards the contribution and novelty of our work, which is different from regular papers that focus on individual techniques. We clarify our contribution from three points:
> >  1. For the (Chinese) industry, at this time of transition from traditional NLP solutions to LLMs, a thorough empirical study to systematically explore the available models, data and efficient methods  is highly anticipated.  When customizing their own LLM models, their need is not for "novelty", but for a "guide book" (findings in **Chapter 2**) that can serve as a useful reference for them to help them know what choices are available in open source community and provide guidance on these (models, data, etc.) choices.  Therefore, our focus is not only pursuing the "insights" and "novelties". Nevertheless, we try our best to analyze the technology behind each choice for detailed reasons.
> >  2. The feasibility of verifying some existing works in Chinese scenarios is still "new insight".  For example, the role of CoT datasets in reasoning  tasks in Chinese scenarios, although there is no technological innovation, has yet to be confirmed. After all, there are significant differences in the basic logic of expression between Chinese and English.  These findings ( in **Chapter 3**) could help generate broader, more universally applicable insights.
> >  3. From the technical perspective, we customized an LLM  that is comparable to chatglm, with only a few parameters trainable. This can help follow-ups customize strong Chinese LLMs in resource-constrained situations.

---

### Official Review · Reviewer_3VDv · 2023-08-02

**Soundness:** 3

**Excitement:**

3: Ambivalent: It has merits (e.g., it reports state-of-the-art results, the idea is nice), but there are key weaknesses (e.g., it describes incremental work), and it can significantly benefit from another round of revision. However, I won't object to accepting it if my co-reviewers champion it.

**Paper Topic And Main Contributions:**

- This paper compares and contrasts a range of open language learning models (LLMs), datasets, and learning methods used for instruction-tuning large language models (LLMs) in Chinese, through extensive experimentation.
- It also explores the effects of additional training when applying LLMs, primarily trained in English, to the Chinese language.
- The authors plan to release a Chinese LLM with performance comparable to ChatGLM.

**Reasons To Accept:**

- The paper provides abundant quantitative comparisons from numerous experimental perspectives, which would be valuable for readers who plan to utilize these models and datasets.
- Although there are minor typos, the content is clearly written and can be easily understood by the reader.
- The application of English LLMs to Chinese provides insights that could be potentially applicable to multiple languages, likely attracting a wide range of readers.

**Reasons To Reject:**

- The majority of the paper is a listing of evaluations using existing datasets and models, with little in the way of new insights. While each experiment includes a brief discussion, many statements lack substantial evidence, resulting in few generalizable insights. Consequently, the knowledge that remains once these models and datasets become obsolete may be limited. The authors should devote more attention to discussions that can yield insights applicable beyond this study, such as the application of English LLMs to Chinese discussed in Chapter 3. This could help to generate broader, more universally applicable insights.
- While the overall content and results are readily comprehensible, there are many deficiencies in the description of existing models and datasets. Much information is only written in table captions when it should be comprehensible from the text alone. Similarly, it seems inappropriate to have statements that rely on results from the appendix.
- All evaluations are conducted with ChatGPT, without any human evaluation. Although the ChatGPT evaluation is somewhat reliable, it has biases, such as assigning low scores to short sentences, as the authors themselves noted. Therefore, there are doubts about whether the results obtained from these experiments should be taken at face value.

**Reproducibility:**

4: Could mostly reproduce the results, but there may be some variation because of sample variance or minor variations in their interpretation of the protocol or method.

**Reviewer Confidence:**

3: Pretty sure, but there's a chance I missed something. Although I have a good feel for this area in general, I did not carefully check the paper's details, e.g., the math, experimental design, or novelty.

---

> ### Author Rebuttal · Authors · 2023-08-26
>
> + **Q1**: "The knowledge that remains once these models and datasets become obsolete may be limited. " & Chapter2's insights?
> > First of all, we fully agree with the suggestion of 'developing more attention to discussions that can yield broader insights applicable'. This is also the goal of our Chapter 3. However, we have different opinions on 'these discussions (in our Chapter 2) on existing datasets and models results in few generalizable insights', as follows:
> > 1. The success of ChatGPT is promoting the gradual transformation of solutions for various NLP tasks in the (Chinese) industry towards LLMs. However, the vast majority of companies are unable to train their own pre-trained models and instead customize their own models based on models and data from the open source community. At this time of transformation, a thorough empirical study to systematically explore the available models and data  is meaningful and highly anticipated by the open source community. Their need is not for 'novelty', but for an "guide book" that can serve as a useful reference for them to help them know what choices are available and provide guidance on these (models, data, etc.) choices when fine-tuning LLM with Chinese instructions. As you mentioned , '... would be valuable for readers who plan to utilize these models and datasets.', therefore, our focus is not only pursuing the "insights" and "novelties".
> >  2. Even though these datasets and models will eventually become obsolete, we believe that our work is still meaningful. Because our analysis focuses more on the technology behind them rather than the data or models themselves, e.g., analyzing the reasons of poor performance of Moss,  comparing the instruction data of self-instruct and nlp-dataset collections, and etc.
> >  3. The feasibility of verifying some existing works in Chinese scenarios is still "new insight". For example, the role of CoT datasets in reasoning  tasks in Chinese scenarios, although there is no technological innovation, has yet to be confirmed. After all, there are significant differences in the basic expression logic between Chinese and English.
>
> + **Q2**: "While the overall content and results are readily comprehensible, there are many deficiencies in the description of existing models and datasets".
> > As you know, our paper is dominated by a large number of experiments and analysis, resulting in a serious shortage of space. We repeatedly streamline and compress space as much as possible without losing information, which leads into the situation that "much information is only written in table captions". We will further improve the presentation of the paper in the next version.
>
> + **Q3**: "All evaluations are conducted with ChatGPT" ?
> > We believe this is a serious misunderstanding and factual error, which may  affect your assessment of our soundness score.
> > 1. Firstly, **not all evaluations in our paper are conducted with ChatGPT**. We only use ChatGPT to evaluate Belle-eval, but for multiple-choice questions (i.e., benchmark  MMCU), scores can be directly computed automatically.
> > 2. Secondly, two benchmarks used in our paper focuses on open generation and closed multi-choice selection, respectively. The former is indeed more suitable for human evaluation. However, it is hard to conduct human evaluation for the hundreds-of-group experiments and millions of samples in our paper. Although ChatGPT has certain biases, it is still the most reliable open generation evaluation method currently.  We think that compared to human evaluation that can only conducted to a small number of samples, ChatGPT, which can be extended to more test samples, may provide a more comprehensive evaluation.
> > 3.  Finally, a very common English benchmark for instruct-tuning LLMs is MMLU. Inspired by this, in the Chinese scenario, we naturally chose an MMCU that is close to MMLU's positioning and construction procedure. We believe this benchmark can also evaluate LLM capabilities reliably. Therefore, based on the comprehensive analysis of the results of both Belle-eval and MMCU, we believe that our assessment is credible.

---

### Official Review · Reviewer_L1A9 · 2023-08-04

**Typos Grammar Style And Presentation Improvements:** 1. line 025
**Soundness:** 3

**Excitement:**

3: Ambivalent: It has merits (e.g., it reports state-of-the-art results, the idea is nice), but there are key weaknesses (e.g., it describes incremental work), and it can significantly benefit from another round of revision. However, I won't object to accepting it if my co-reviewers champion it.

**Paper Topic And Main Contributions:**

Chinese LLMs are still in the early stage. In order to provide valuable findings for effectively customizing LLMs to better respond to Chinese instructions, this work makes an in-depth empirical study of instruction-tuning LLMs in Chinese from three most important aspects - LLM bases, parameter-efficient methods, and instruction data types, as well as other factors like COT and human-value alignment. Moreover, this paper will release a powerful Chinese LLMs comparable to ChatGLM.

**Reasons To Accept:**

This work is the first to systematically study on instruction-tuning in Chinese. The authors have conducted intensive experiments to find the best combination of LLM bases, parameter-efficient methods and instruction datasets, as well as other factors. I think these findings are very valuable for researchers to develop better Chinese LLMs.

**Reasons To Reject:**

1. Because of the variables controlling method used in the paper, the "best" combination of factors is only locally optimal. And the results depend largely on the initially selected parameter-efficient methods and instruction datasets.
2. The reasons that selecting Belle-eval and MMCU as two benchmarks are not very convincing. And one point that is very confusing to me is that the choice of instruction datasets may have a great impact on the evaluation results. That is, the more similar the data distribution between instruction datasets and the benchmark, the higher performance the model will obtain.

**Reproducibility:**

4: Could mostly reproduce the results, but there may be some variation because of sample variance or minor variations in their interpretation of the protocol or method.

**Reviewer Confidence:**

4: Quite sure. I tried to check the important points carefully. It's unlikely, though conceivable, that I missed something that should affect my ratings.

---

> ### Author Rebuttal · Authors · 2023-08-26
>
> We sincerely thank you for the agreement, thorough review and constructive comments.
>
> +  **Q1**: the "best" combination of factors is only locally optimal."
> > We fully understand the concern of "locally optimal", since we did not traverse all the three-factor combinations (LLMs, data, parameter-efficient methods), but instead controlled two variables to traverse all possibilities of the current variable. We will respond to this consideration from two aspects:
> > 1.  If we simultaneously traverse all combinations of the three factors, it will result in hundreds of combinations, which requires high computational costs, and we do not think that there will be more valuable findings besides searching the "best" combination.
> > 2. The main focus of this paper is to conduct a thorough analysis of each factor, such as the impact of different instruction construction methods, and the Chinese capabilities of different LLMs. These findings may have persistent reference value for follow-ups to customize their own Chinese LLMs, which is more urgently needed for Open-source community than simply searching a "best" combination.
>
> + **Q2**: "The more similar the data distribution between instruction datasets and the benchmark, the higher performance the model will obtain." & Reasons to select the two benchmark.
> > We fully agree with this concern, but in fact, any field or task will face the same problem of "training on datasets closer to the distribution of the test set can achieve better results".  It is precisely with this in mind that our paper chose the two benchmarks, which have significant distribution differences to avoid the impact of data distribution.
> >  Specifically, most of the former are general instructions that require LLM to generate open and **longer** replies. The latter are mostly instructions in the form of multiple choice questions, requiring LLMs output (**shorter**) options. The two represent the capabilities of **open** generation and **closed** set selection, and it is difficult for any dataset to simultaneously approximate these two vastly different benchmarks. Therefore, we believe that our evaluation is persuasive.
> >  In addition, more experiments in this paper are not affected by this concern, such as comparing models, vocabulary, prompt languages, etc., **under the same instruction dataset**. These experiments also have high reference value and broader applicable insights.
>
> + **Q3**: Some typos and presentation improvements:
> > The typos have be properly addressed as we reply. Besides, we have optimized the Fig. 1 and tables to make them more reader friendly.

---

### Meta-Review · Area_Chair_uMsr · 2023-09-16

**Recommendation:** 3

**Metareview:**

This paper conducts extensive experiments to compare a range of LLMs, datasets, and learning methods used for instruction-tuning LLMs in Chinese and investigates other factors including vocabulary and prompt language. The reviewers in general think the paper is well-written and is valuable for practitioners who plan to utilize these models and datasets, but feel less excited about it. I'd recommend accepting it into Findings.

---

### Decision · Program_Chairs · 2023-10-07

**Decision:**

Accept-Findings

**Comment:**

This paper conducts extensive experiments to compare a range of LLMs, datasets, and learning methods used for instruction-tuning LLMs in Chinese and investigates other factors including vocabulary and prompt language. The reviewers in general think the paper is well-written and is valuable for practitioners who plan to utilize these models and datasets, but feel less excited about it. I'd recommend accepting it into Findings.